# Fatigue and Crack Growth under Constant- and Variable-Amplitude Loading in 9310 Steel Using "Rainflow-on-the-Fly" Methodology

**James C. Newman, Jr.**

Department of Aerospace Engineering, Mississippi State University, Mississippi, MS 39762, USA; newmanjr@ae.msstate.edu; Tel.: +1-(901)-734-6642

**Abstract:** Fatigue of materials, like alloys, is basically fatigue-crack growth in small cracks nucleating and growing from micro-structural features, such as inclusions and voids, or at micro-machining marks, and large cracks growing to failure. Thus, the traditional fatigue-crack nucleation stage ($N_i$) is basically the growth in microcracks (initial flaw sizes of 1 to 30 μm growing to about 250 μm) in metal alloys. Fatigue and crack-growth tests were conducted on a 9310 steel under laboratory air and room temperature conditions. Large-crack-growth-rate data were obtained from compact, C(T), specimens over a wide range in rates from threshold to fracture for load ratios (R) of 0.1 to 0.95. New test procedures based on compression pre-cracking were used in the near-threshold regime because the current ASTM test method (load shedding) has been shown to cause load-history effects with elevated thresholds and slower rates than steady-state behavior under constant-amplitude loading. High load-ratio (R) data were used to approximate small-crack-growth-rate behavior. A crack-closure model, FASTRAN, was used to develop the baseline crack-growth-rate curve. Fatigue tests were conducted on single-edge-notch-bend, SEN(B), specimens under both constant-amplitude and a Cold-Turbistan+ spectrum loading. Under spectrum loading, the model used a "Rainflow-on-the-Fly" subroutine to account for crack-growth damage. Test results were compared to fatigue-life calculations made under constant-amplitude loading to establish the initial microstructural flaw size and predictions made under spectrum loading from the FASTRAN code using the same micro-structural, semi-circular, surface-flaw size (6-μm). Thus, the model is a unified fatigue approach, from crack nucleation (small-crack growth) and large-crack growth to failure using fracture mechanics principles. The model was validated for both fatigue and crack-growth predictions. In general, predictions agreed well with the test data.

**Keywords:** fatigue; crack growth; metallic materials; plasticity; crack closure; spectrum loading

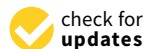



## 1. Introduction

This article is dedicated to Dr. Wolf Elber and his remarkable achievements.

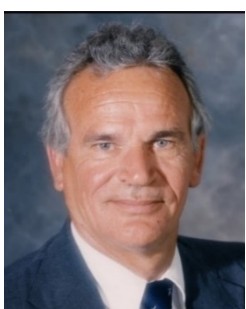

On 12 January 2019, Wolf took off on his final flight. His passion was to soar in his glider over the Blue Ridge Mountains in southwest Virginia, USA. With his passing,

the fatigue and fracture mechanics community lost a great pioneer. Dr. Elber had a distinguished career at the NASA Langley Research Center, and later as head of the U.S. Army Research Laboratory at Langley. His professional accolades are immortalized by his discovery of the "plasticity induced crack closure" phenomenon [1,2]. Thank you, Wolf, for your friendship and brilliant mind over the years. We will greatly miss you.

Fatigue of metallic materials is divided into several phases: crack nucleation, small- and large-crack growth, and fracture [3]. Crack nucleation is controlled by local stress and strain concentrations and is associated with cyclic slip-band formation from dislocation movement leading to intrusions and extrusions [4,5]. Although cyclic slip may be necessary in pure metals, the presence of inclusions, voids, or machining marks in metal alloys greatly affects the crack-nucleation process. (Herein, small-crack growth includes microstructurally and physically small cracks.) The small-crack growth regime is the growth in cracks from inclusions, voids, or machining marks, ranging from 1 to 30 μm in depth [6]. Schijve [7] has shown that, for polished surfaces of pure metals and commercial alloys, the formation of a small crack of about 100 μm in size can consume 60 to 80% of the fatigue life. This is the reason that there is so much interest in the growth behavior of small cracks. Large-crack growth and failure are regions where fracture-mechanics parameters have been successful in correlating and predicting fatigue-crack growth and fracture. In the past three decades, fracture-mechanics concepts have also been successful in predicting the growth in small cracks under constant-amplitude and spectrum-loading using crack-closure theory [6].

The engineered metallic materials are inhomogeneous and anisotropic when viewed on a microscopic scale. For example, these materials are composed of an aggregate of small grains, inclusion particles or voids. These inclusion particles are of different chemical compositions to the bulk material, such as silicate or alumina inclusions in steels. Because of their nonuniform microstructure, local stresses may be concentrated at these locations and may cause the initiation of fatigue cracks. Crack initiation is primarily a surface phenomenon because: (1) local stresses are usually highest at the surface, (2) an inclusion particle of the same size has a higher stress concentration at the surface than in the interior, (3) the surface is subjected to adverse environmental conditions, and (4) the surfaces are susceptible to inadvertent damage. The growth in "naturally" initiated cracks in commercial aluminum alloys has been investigated by Bowles and Schijve [8], Morris et al. [9] and Kung and Fine [10]. In some cases, small cracks initiated at inclusions and the Stage I period of crack growth were eliminated [6]. This tendency toward inclusion initiation rather than slip-band (Stage I) cracking was found to depend on stress level and inclusion content [10]. Similarly, defects (such as tool marks, scratches and burrs) from manufacturing and service-induced events will also promote initiation and Stage II crack growth [6].

During the last three decades, test and analysis programs on "small-crack" behavior have shown that majority of the fatigue life is consumed by small-crack growth from a micro-structural feature for a variety of metal alloys [11–14]. The smallest measured flaw sizes using plastic-replica methods ranged from 10 to 30 μm for aluminum alloys (2024-T3; 7075-T6), aluminum–lithium alloy (2090-T8ED41) and 4340 steel; and the crack-propagation life was about 90 percent of the fatigue life for constant-amplitude and spectrum loading. Thus, a large portion of nucleation life ($N_i$) in classical fatigue is small-crack growth, from a micro-structural feature to about 250 μm for these materials. The exception was the Ti-6Al-4V alloy [13], where the smallest measured flaw sizes were about 20 μm and crack-growth lives were about 50% of the fatigue life. However, small-crack analyses of similar Ti-6Al-4V specimens machined from two engine discs [15,16] showed that the initial flaw sizes from 2 to 20 μm predicted the scatter band in fatigue life on open-hole fatigue tests quite well. Therefore, the crack-growth approach provides a unified theory for the determination of fatigue lives for these materials. However, for pure- and single-crystal materials, nucleation cycles are required to transport dislocations at critical locations, develop slip bands, and cracks.

Fatigue-crack growth under variable-amplitude and spectrum loading is composed of complex crack-shielding mechanisms (plasticity, roughness and fretting debris) and damage accumulation due to cyclic plastic deformations around a crack front in metallic materials. Typically, "rainflow" methods [17] are applied to variable-amplitude loading to develop a load sequence that is used to compute damage accumulation and life because damage relations are, generally, a non-linear function of the crack-driving parameters. However, crack-tip damage is only a function of the current loading and load history (material memory). Loading in the future has no bearing on the "current" damage. The life-prediction code, FASTRAN [18], has a "rainflow-on-the-fly" methodology [19] to compute damage as the cyclic load history is applied, and there was no need to reorder the spectra. Herein, several variable-amplitude loading sequences were developed to test and to validate the "rainflow-on-the-fly" subroutine.

The paper presents the results of large-crack-growth-rate tests conducted on compact, C(T), specimens made of 9310 steel [20] over a wide range of constant-amplitude loading to establish the baseline crack-growth-rate curve for fatigue and crack-growth analyses. Compression pre-cracking methods [21–26] were used to generate test data in the near-threshold regime because the ASTM E-647 test method [27,28] using the load-shedding procedure was shown to cause load-history effects and slower crack-growth rates than steady-state behavior [25,26]. Both compression pre-cracking constant-amplitude (CPCA) and load-reduction (CPLR) methods were used. A crack-closure analysis was used to collapse the rate data from C(T) specimens into a narrow band over many orders of magnitude in rates using a plane-strain constraint factor for low rates and modeled a constraint-loss regime to plane-stress behavior at high rates. For steels, small- and large-crack data (without load-history effects) tend to agree well [29]. Thus, the high-R large-crack data in the near-threshold regime is a good estimate for small-crack behavior, as proposed by Herman et al. [30]. A Two-Parameter Fracture Criterion [31] was used to characterize the fracture behavior. Fatigue tests were conducted on 9310 steel single-edge-notch-bend, SEN(B), specimens [32] under both constant-amplitude and a modified Cold-Turbistan [33] spectrum loading. The test results were compared to the life calculations or predictions made using the FASTRAN code.

## 2. Material and Specimen Configurations

The Boeing Company (Seattle, WA, USA) provided a 9310 steel rod (150 mm diameter by 950 mm length) to Mississippi State University. C(T), SEN(B), and tensile specimens ($B$ = 6.35 mm) were machined in the longitudinal direction (cracks perpendicular to longitudinal direction) and heat-treated by special procedures [20]. The yield stress, $\sigma_{ys}$, was 980 MPa, ultimate tensile strength, $\sigma_u$, was 1250 MPa, and modulus of elasticity was $E$ = 208.6 GPa. Figure 1 shows C(T) and SEN(B) specimens with back-face strain (BFS) gauges used to monitor crack growth in the C(T) specimens and crack nucleation and growth in the SEN(B) specimen.

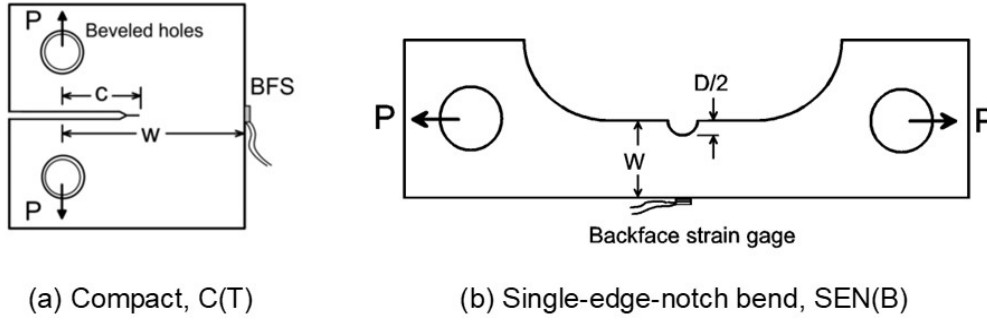

(a) Compact, C(T)                    (b) Single-edge-notch bend, SEN(B)

**Figure 1.** Fatigue-crack-growth and fatigue specimens tested and analyzed.

### 3. "Rainflow-on-the-Fly" Methodology

Since the Paris–Elber non-linear power law is used to compute crack-growth damage in the FASTRAN life-prediction code [18], a "rainflow-on-the-fly" subroutine was developed to compute crack growth under variable-amplitude loading. Damage calculations are illustrated in Figure 2 and they are based on calculations of stress amplitudes above the crack-opening stress, $S_o$. Damage only occurs during the loading amplitude but unloading may change crack-tip deformations and affect the subsequent damage during the next loading amplitude. Crack-opening stress, $S_o$, is calculated at the minimum stress, $S_1$, and $\Delta c_1$ is a function of $\Delta S_{eff} = S_2 - S_o$. The $\Delta S_{eff}$ value is used to compute $\Delta K_{eff}$ and then the crack-growth rate ($dc/dN$) per loading amplitude, which gives $\Delta c_1$. The next unloading to $S_3$ did not close the crack, and the next maximum loading was to $S_4$. Here, damage is $\Delta c_2 = f(S_4 - S_o) - \Delta c_1 + f(S_2 - S_3)$, which captures the larger damage due to $S_4$. The total damage is $\Delta c = \Delta c_1 + \Delta c_2$. Again, each stress range is used to calculate an effective stress-intensity factor and then the corresponding crack extension from the crack-growth-rate relation. Application of minimum stress, $S_5$, caused the crack surfaces to close, and rainflow-on-the-fly logic was reset. Total damage, $\Delta c$, captures the essence of rainflow logic and applies damage in the proper sequence using a cycle-by-cycle calculation. (Note that a cycle is defined as any minimum–maximum–minimum stress.)

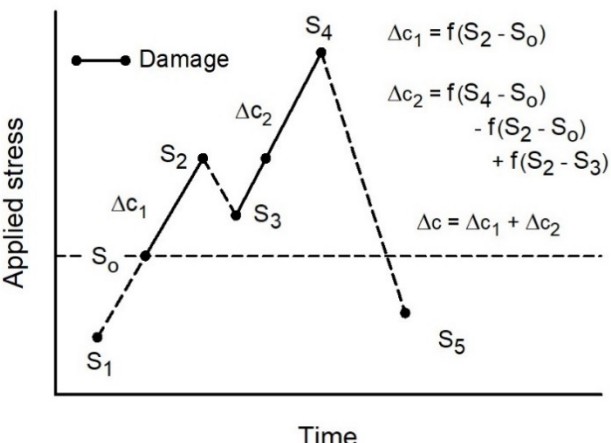

**Figure 2.** Damage calculations using crack-closure theory under variable-amplitude loading.

Herein, several specially designed spectra were developed to exercise the FASTRAN code and calculations are shown for these spectra. Each spectrum had sequences of stress ranges that would require "rainflow" logic to calculate the proper crack extension. Comparisons are made between: (1) linear damage, (2) rainflow logic and (3) FASTRAN. Linear damage is using only the stress range to calculate crack extension and ignoring load interactions. Rainflow logic is a manual calculation of crack extension using the basic rainflow concept principles to compute the proper crack extension. FASTRAN uses the "rainflow-on-the-fly" subroutine to remember stress history and to compute the proper damage as the crack-opening stress in FASTRAN was held constant, as specified, and the crack-growth relation was a simple Paris–Elber relation as

$$dc/dN = 5 \times 10^{-10} (\Delta K_{eff})^3 \tag{1}$$

where $\Delta K_{eff} = \Delta S_{eff} \sqrt{\pi c}$. The spectra were applied to an infinite plate under remote uniform stress, $S$, with an initial crack half-length ($c_i$) of 5 mm.

Spectrum A is a Christmas-Tree type loading sequence and is shown in Figure 3a and was designed to severely test the Rainflow-on-the-Fly option in FASTRAN. Without Rainflow analyses, the difference in crack-growth lives for the sequence would be several orders-of-magnitude in error using linear damage [19]. One block of loading is defined from point A to B, and the sequence was repeated until the final crack length was reached

or the specimen failed. The analytical crack-closure model in FASTRAN was turned off by intentionally setting the crack-opening stress, $S_o$, to a constant value of 50 MPa.

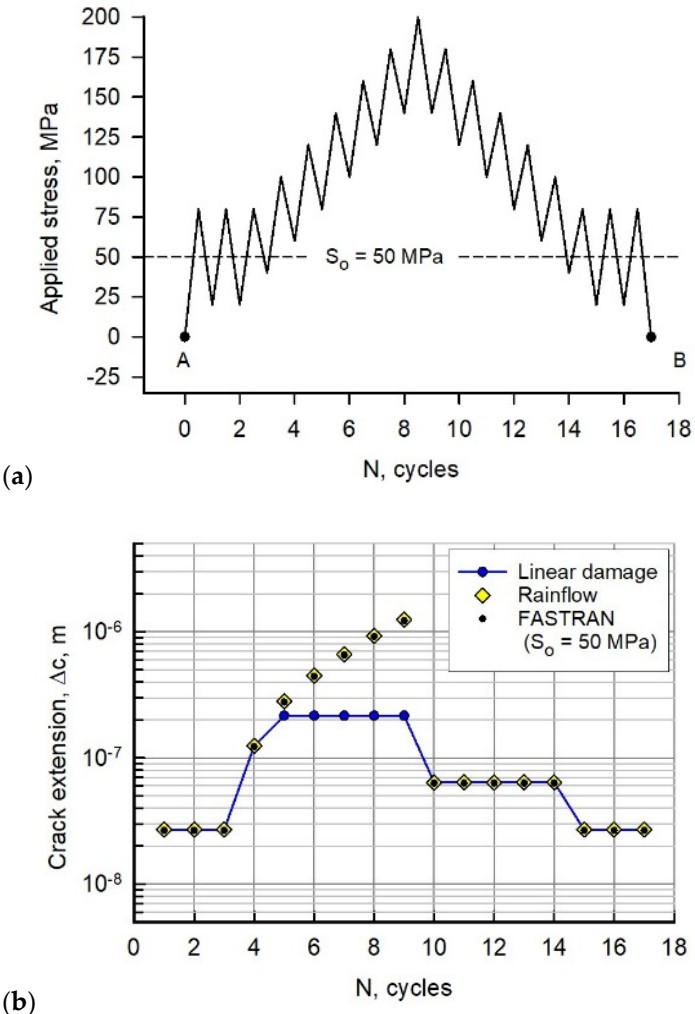

**Figure 3.** (**a**) Loading sequence A. (**b**) Calculated crack extension per applied cycle for load sequence A.

Figure 3b shows crack-growth increments calculated for each loading amplitude. If the crack-opening stress is greater than the minimum stress, $S_{min}$, then $\Delta S_{eff} = S_{max} - S_o$. If $S_{min}$ is greater than the crack-opening stress, then $\Delta S_{eff} = S_{max} - S_{min}$. These equations apply only for "linear damage" (lines with a solid blue symbol) and for loadings that do not require Rainflow methods. Diamond symbols show what a Rainflow method would give in crack extension. (Note that a cycle is defined as any minimum to maximum to minimum loading, and crack-growth damage only occurs during loading. The unloading part of the cycle causes reverse plastic deformations that may affect damage during the next load application.) FASTRAN with the Rainflow subroutine gives the solid (black) symbols that exactly match what the Rainflow method predicted.

The second spectrum (B) was designed after some of the European standard spectra with various stress amplitudes that have a number of constant-amplitude cycles, and the spectrum had a number of cycles that go from maximum–minimum–maximum or minimum–maximum–minimum loading. Some cycles had a Christmas-Tree type loading that was interrupted with other stress amplitudes. Again, this spectrum does require rainflow logic to calculate the correct damage.

The first case (B1) had a constant crack-opening stress (50 MPa) for the complete spectrum, as shown in Figure 4a. The calculated crack extension using linear damage, rainflow and FASTRAN are shown in Figure 4b. Out of the 57 cycles, 18 cycles required

rainflow logic to compute the correct crack extension. FASTRAN agreed with the Rainflow predicted crack extensions.

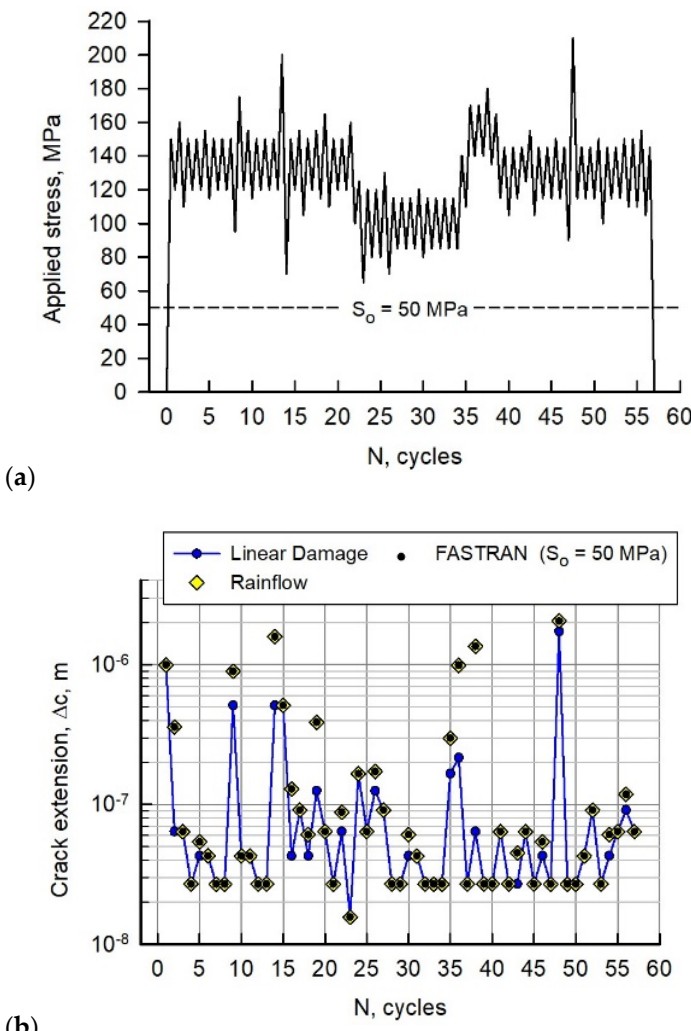

(**a**)

(**b**)

**Figure 4.** (**a**) Loading sequence B1. (**b**) Calculated crack extension per applied cycle for load sequence B1.

In the FASTRAN code, crack-opening stresses will change as a function of stress history. Thus, to simulate a changing crack-opening stress, the second case (B2) had a constant crack-opening stress of 50 MPa for the first 22 cycles and then $S_o$ = 90 MPa for the remainder of loading, as shown in Figure 5a. Of course, the first 22 cycles have the same behavior as Case B1, but now cycles 23 to 34 do not require rainflow logic ($S_{min} < S_o$) and crack extensions were computed directly from the Paris–Elber relation (Equation (1)). These crack-extension comparisons are shown in Figure 5b. However, the crack extensions during cycles 35 to 37 were much less than in Case B1 due to the higher crack-opening stress and lower $\Delta K_{eff}$ values.

Figure 6 shows the calculated crack-opening stresses from FASTRAN ($\alpha$ = 2) using the full crack-closure model for load sequence B using the cycle-by-cycle option. The results for Block 1 are shown as the dashed (red) lines. The crack-opening stresses are calculated at a minimum applied stress and remain constant until the crack closes again at the next minimum applied stress. In many cycles, the crack-tip region was open ($S_{min} > S_o$) during the applied stress amplitudes. The model starts with a "zero" crack-opening stress and the opening stresses increase as the crack grows and leaves plastically deformed material in the wake of the crack tip.

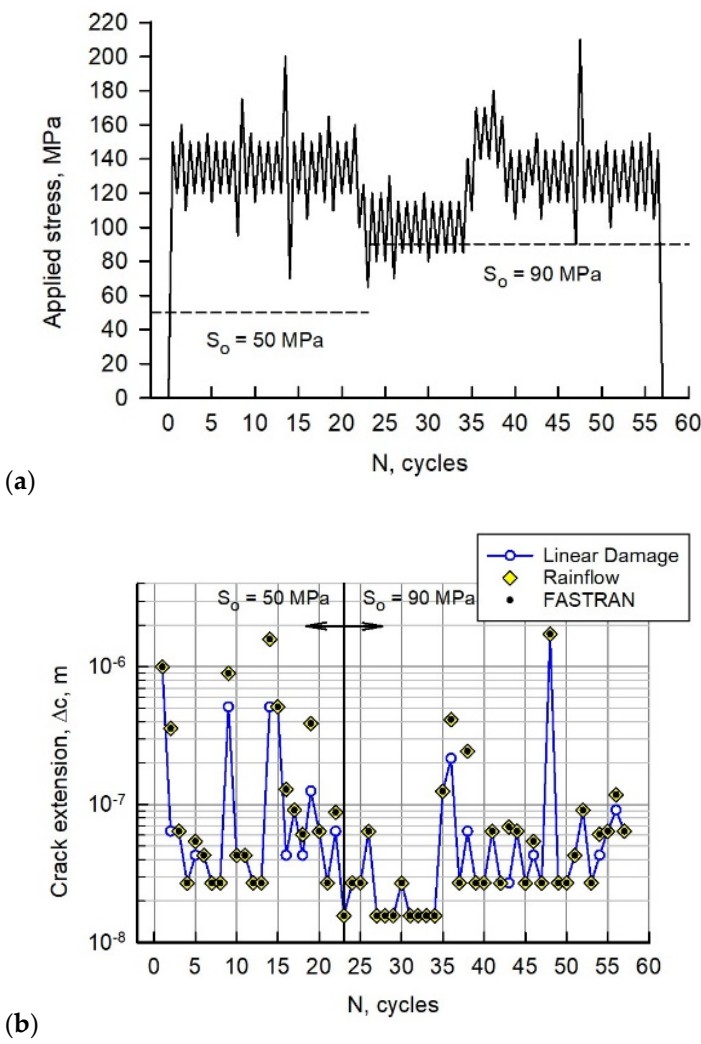

(**a**)

(**b**)

**Figure 5.** (**a**) Loading sequence B. (**b**) Calculated crack extension per applied cycle for load sequence B2.

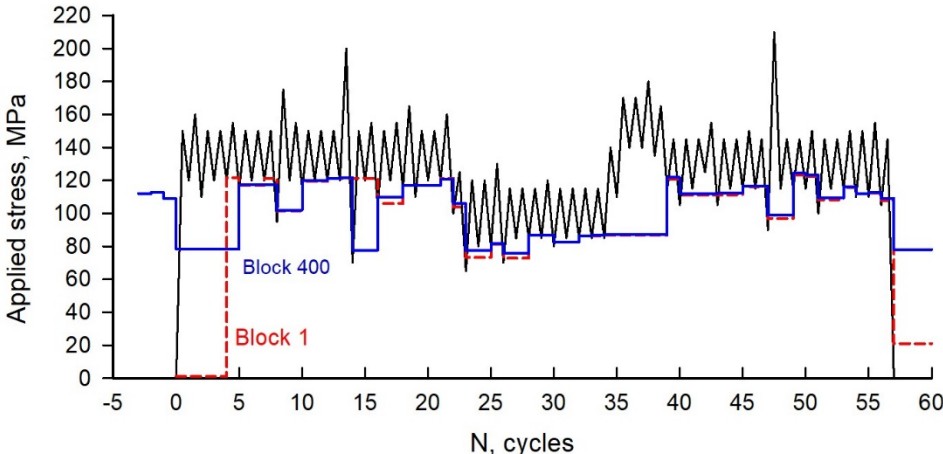

**Figure 6.** Loading sequence B and calculated crack-opening stresses.

Calculated crack-opening-stress history for Block 400, at about 110,000 cycles (half-life), are shown as solid (blue) lines. Here, the cycle numbers are the same cycles as in Block 1 (first sequence). The calculated crack-opening stresses where in agreement over most of the sequence, except the only major difference was in cycles 1 to 5, where the crack surfaces were fully open. However, the crack extension during the first five cycles in Block 1 was

a factor of 8 larger than that during the 400th block because of the higher crack-opening stress. Most of the damage from cycles 1 to 5 was from the loading on cycle 1.

## 4. Fatigue-Crack-Growth and Fracture Tests on 9310 Steel

Compact, C(T), specimens were used to generate the $\Delta K$ against rate ($dc/dN$) data on the 9310 steel at room temperature and 20 Hertz [20] over a wide range in stress ratios ($R$ = 0.1 to 0.95). These $\Delta K$-rate data are shown in Figure 7. Tests were conducted from near threshold to fracture. A BFS gauge was used to monitor crack growth and to measure crack-opening loads using the compliance-offset method. In the low-rate regime, compression pre-cracking constant amplitude (CPCA) and load reduction (CPLR) methods were used to generate the $\Delta K$-rate data. In general, the test frequency was dropped to about 5 to 8 Hertz as the cracks were grown to failure. The data show the normal spread with the R value but shows more spread in the threshold and fracture regions. In the fracture region, the spread is due to the fracture being controlled by the maximum stress-intensity factor and would not collapse on a $\Delta K$-rate plot. For example, the stress-intensity-factor range at failure, $\Delta K_c = K_{Ie} (1 - R)$, where $K_{Ie}$ is the elastic fracture toughness. Thus, high $R$ tests would fracture at a lower $\Delta K_c$ value than low $R$ tests. The spread in the threshold region is suspected to be due to the load-history effects caused by the load-reduction procedure used in the CPLR method. Load-reduction tests, as specified in ASTM E-647 [28], have been shown [25,26] to induce more spread in $\Delta K$ in the threshold region than the mid-rate region. The spread has been associated with a rise in the crack-closure behavior during load-reduction tests [34,35].

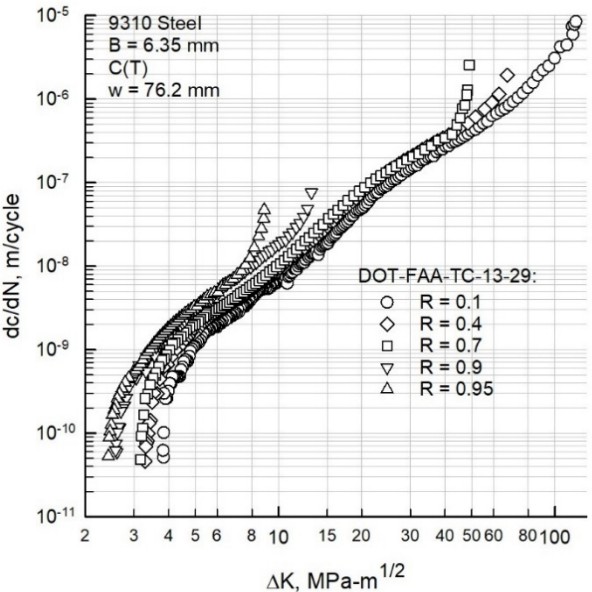

**Figure 7.** Stress-intensity-factor range against rate for 9310 steel over range in stress ratio (R).

To evaluate the fracture toughness of the material, only the fatigue-crack-growth tests were available. In most tests, the cracks in the C(T) specimens were grown to failure under cyclic loading. Here, the final recorded crack length and the maximum fatigue load were used to calculate $K_{Ie}$ (elastic stress-intensity factor at failure), and these results are shown in Figure 8, as solid (blue) circular symbols. Most tests failed at large $c_i/w$ ratios (>0.7). For comparison, similar test results conducted on a 4340 steel [36] are shown as open circular symbols. Here, one 4340 steel specimen was fatigue cracked to a lower crack-length-to-width ratio and statically pulled to failure (solid square symbol).

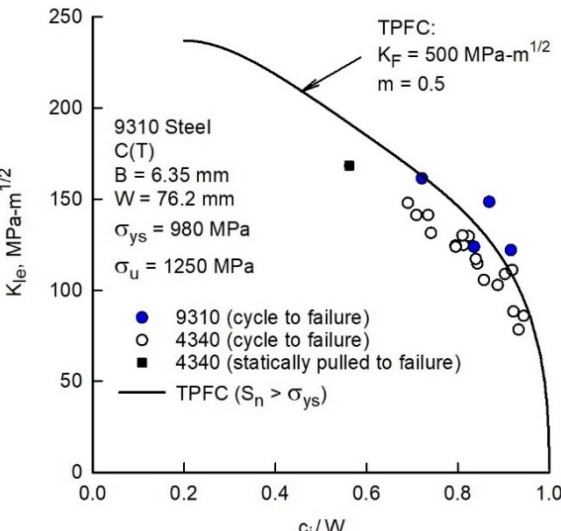

**Figure 8.** Elastic stress-intensity factor at failure on 9310 and 4340 steel C(T) specimens.

The solid curve in Figure 8 was based on the Two-Parameter Fracture Criterion (TPFC). A fracture criterion was derived [37] that accounted for the elastic-plastic behavior of a cracked material based on notch-strength analysis. The criterion was based on expressing the elastic stress-intensity factor at failure in terms of net-section stress as

$$K_{Ie} = P_f / \sqrt{BW} \ F = S_n \sqrt{\pi c} \ F_n \tag{2}$$

where $F_n$ is the usual boundary-correction factor based on net-section stress. Equations for $S_n$ and $F_n$ are given in [37] for the C(T) specimen. The fracture criterion is

$$K_F = K_{Ie} / \Phi \tag{3}$$

$$\Phi = 1 - m \, S_n / S_u \tag{4}$$

$$\Phi = (\sigma_{ys} / S_n)(1 - m \, S_n / S_u) \tag{5}$$

where $K_F$ and m are the two material fracture parameters. The stress $S_u$ (ultimate value of elastic net-section stress) was computed from the load required to produce a fully plastic region or hinge on the net section based on the ultimate tensile strength, $\sigma_u$. For the compact specimen, $S_u$ is a function of load eccentricity and is $1.62\sigma_u$ for c/w = 0.5 [37]. The fracture parameters, $K_F$ and $m$, are assumed to be constant in the same sense as the ultimate tensile strength; that is, the parameters may vary with material thickness, state of stress, temperature, and rate of loading. If $m$ is equal to zero in Equations (3) and (4), then $K_F$ is equal to the elastic stress-intensity factor at failure, and the equation represents the behavior of low-toughness (brittle) materials under plane-strain behavior. If $m$ is equal to unity, the equation represents fracture behavior of high-toughness materials (plane-stress fracture).

For given fracture-toughness parameters, $K_F$, and $m$, the elastic stress-intensity factor at failure is

$$K_{Ie} = K_F / (1 + 2m\gamma / \sigma_{ys}) \tag{6}$$

$$K_{Ie} = \left\{ \sqrt{(m\gamma)^2 + 2\gamma S_u} - m\gamma \right\} \sqrt{\pi c} \ F_n \tag{7}$$

$$K_{Ie} = S_u \sqrt{\pi c} \ F_n \tag{8}$$

where $\gamma = K_F \sigma_{ys} / \left[ 2\, S_u \sqrt{\pi c}\ F_n \right]$. In 1973, a relationship between $K_F/E$ and $m$ [37] was found for many materials and crack configurations. Herein, the relationship was used to help select an m value. As the 9310 steel exhibited a high toughness, an $m$-value of 0.5 was selected. The corresponding $K_F$ value was 500 MPa-$\sqrt{}$m to fit the 9310 steel fracture data. Solid curve in Figure 8 was calculated from the TPFC for the 76.2 mm wide compact specimens. These calculations show how the initial crack length affects the elastic fracture toughness. In addition, the variation in specimen width (not shown) would greatly affect the elastic fracture toughness, in that larger width specimens produce large $K_{Ie}$ values.

## 5. Fatigue-Crack-Growth and Crack-Closure Analyses

Crack-opening-stress equations for constant-amplitude loading were developed from an early analytical crack-closure model ($S_o$) calculations [38]. As the number of elements within the plastic-zone region in the model was increased to 20 [18] and the crack-growth increment was modelled on a cycle-by-cyclic basis, new $S_o$ equations were made for a single crack in a very wide plate under uniform remote applied stress, $S$. The new set of equations was developed to fit the results from the revised closure model and, again, gave $S_o$ as a function of stress ratio ($R$), maximum stress level ($S_{max}/\sigma_o$) and the constraint factor ($\alpha$). The new equations are

$$S_o / S_{max} = A_0 + A_1 R + A_2 R^2 + A_3 R^3$$

(9)

$$S_o / S_{max} = A_0 + A_1 R$$

(10)

where $R = S_{min}/S_{max}$, $S_{max} < 0.8\ \sigma_o$, and $S_{min} > -\sigma_o$. The $A_i$ coefficients are functions of $\alpha$ and $S_{max}/\sigma_o$ and are given by

$$A_0 = \left( 0.9453 - 0.514\,\alpha + 0.1355\,\alpha^2 - 0.0133\,\alpha^3 \right) \left[ \cos(\beta) \right]^{(0.8\,\alpha - 0.1)}$$

(11)

$$\beta = \pi S_{max} / (2\alpha\sigma_o)$$

$$A_1 \left( = 0.5719 - 0.1726\,\alpha + 0.019\,\alpha^2 \right) S_{max} / \sigma_o$$

(12)

$$A_2 = 0.975 - A_0 - A_1 - A_3$$

(13)

$$A_3 = 2A_0 + A_1 - 1$$

(14)

A crack-closure analysis was then performed on the fatigue-crack growth ($\Delta K$-rate) data from C(T) specimens in Figure 7 to determine the $\Delta K_{eff}$-rate relation. The $K$-analogy concept [18] was used to calculate the crack-opening stresses (or loads) for C(T) specimens from the above equations. The $\Delta K_{eff}$-rate data are shown in Figure 9. Selection of the lower constraint factor, 2.5, was found to reasonably collapse the $\Delta K$-rate data into an almost unique relation.

In the threshold region, the lower R tests exhibited a rise in crack-opening loads as the $\Delta K$ level was reduced in a load-reduction test. Even the CPLR method showed a load-history effect, but not as much as the current ASTM procedure [28]. The upper constraint factor, 1.15, and constraint-loss range was selected to help fit spectrum crack-growth tests [20]. The lower vertical dashed line at $(\Delta K_{eff})_{th}$ is the estimated threshold for the steel [39], and the upper vertical dashed line at $(\Delta K_{eff})_T$ is the location of constraint loss from plane-strain to plane-stress behavior [40]. The solid (blue) lines with circular (yellow) symbols shows the baseline crack-growth-rate curve for FASTRAN. Fatigue-crack-growth, fracture and tensile properties for the 6.35-mm thick 9310 steel are given in Table 1.

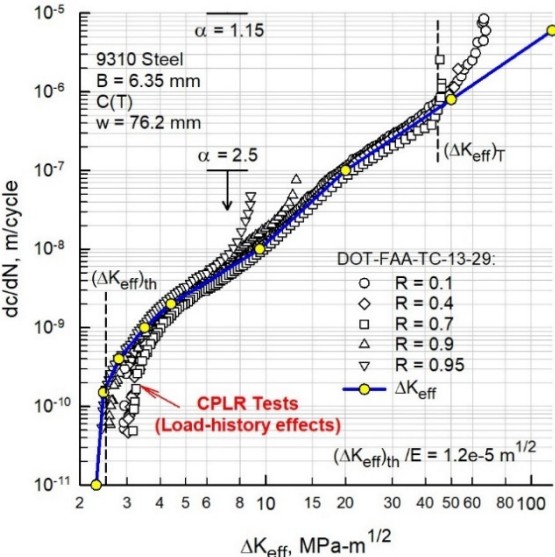

**Figure 9.** Effective stress-intensity factor range against rate for 9310 steel C(T) specimens.

**Table 1.** Effective stress-intensity factor range against rate, fracture and tensile properties for 9310 steel ($B$ = 6.35 mm).

| $\Delta K_{eff}$, (MPa-m$^{1/2}$) | $da/dN$ and $dc/dN$ (m/cycle) | Crack-Growth, Fracture and Tensile Properties |
|:---:|:---:|:---:|
| 2.30 | $1.0 \times 10^{-11}$ | $\Delta K_o = 0$ $q = 6$ |
| 2.45 | $1.5 \times 10^{-10}$ | $\alpha_1 = 2.5$ at $1.0 \times 10^{-7}$ m/cycle |
| 2.80 | $4.0 \times 10^{-10}$ | $\alpha_2 = 1.15$ at $1.0 \times 10^{-5}$ m/cycle |
| 3.50 | $1.0 \times 10^{-9}$ | $K_F = 500$ MPa-m$^{1/2}$ |
| 4.40 | $2.0 \times 10^{-9}$ | $m = 0.5$ |
| 9.50 | $1.0 \times 10^{-8}$ | $\sigma_{ys} = 980$ MPa |
| 20.0 | $1.0 \times 10^{-7}$ | $\sigma_u = 1250$ MPa |
| 50.0 | $8.0 \times 10^{-7}$ | $\sigma_o = 1115$ MPa |
| 120.0 | $6.0 \times 10^{-6}$ | $E = 208.6$ GPa |

## 6. Fatigue Behavior of Notched Specimens

For most fatigue-crack-growth analyses, linear-elastic analyses have been found to be adequate. The linear-elastic effective stress-intensity factor range developed by Elber [2] is given by

$$\Delta K_{eff} = (S_{max} - S_o) \sqrt{\pi c}\, F(c/w) \tag{15}$$

where $S_{max}$ is maximum stress, $S_o$ is crack-opening stress, and $F(c/w)$ is the boundary correction factor. However, for high-stress intensity factors and low-cycle fatigue conditions, linear-elastic analyses are inadequate and nonlinear crack-growth parameters are needed. To account for plasticity, a portion of the Dugdale [41] cyclic-plastic-zone size ($\omega$) has been added to the crack length, $c$. The cyclic-plastic-zone-corrected effective stress-intensity factor [42,43] is

$$(\Delta K_p)_{eff} = (S_{max} - S_o) \sqrt{\pi d}\, F(d/w) \tag{16}$$

where $d = c + \omega/4$ and $F(d/w)$ is the cyclic-plastic-zone-corrected boundary-correction factor. The cyclic plastic zone is given by

$$\omega = (1 - R_{eff})^2\, \rho/4 \tag{17}$$

where $R_{eff} = S_o/S_{max}$ and plastic-zone size ($\rho$) for a crack in a large plate [41] is

$$\rho = c\,\{\sec[\pi S_{max}/(2\alpha\sigma_o)] - 1\} \tag{18}$$

where flow stress, $\sigma_o$, is multiplied by the constraint factor ($\alpha$). Herein, the cyclic-plastic-zone corrected effective stress-intensity factor range (Equation (17)) will be used in the fatigue-life predictions.

The FASTRAN life-prediction code [18] was used to model crack growth from an initial micro-structural flaw size to failure and the crack-growth relation used is

$$dc/dN = C_{1i}[(\Delta K_p)_{eff}]^{C_{2i}} \left\{1 - [\Delta K_o/(\Delta K_p)_{eff}]^p\right\}/[1 - (K_{max}/K_{Ie})^q] \qquad (19)$$

where $C_{1i}$ and $C_{2i}$ are coefficient and exponent for each linear segment ($i$ = 1 to n), respectively. The $(\Delta K_p)_{eff}$ is cyclic-plastic-zone corrected effective stress-intensity factor, $\Delta K_o$ is effective threshold, $K_{max}$ is maximum stress-intensity factor, $K_{Ie}$ is elastic fracture toughness (which is, generally, a function of crack length, specimen width, and specimen type), $p$ and $q$ are constants selected to fit test data in either the threshold or fracture regimes, respectively. Herein, no threshold was modeled and $\Delta K_o$ was set equal to zero; thus, p was not needed. Near-threshold behavior was modeled with the multi-linear equation (independent of $R$). Fracture was modeled using the TPFC ($K_F$ and $m$) [31,37].

Fatigue tests were conducted on SEN(B) specimens under: (1) constant-amplitude loading ($R$ = 0.1) and (2) Cold-Turbistan+ loading. The semi-circular edge notch was chemically polished. The Cold-Turbistan+ spectrum was obtained from [33], adding a constant load to make the overall $R$ = 0.1 (compression loads not allowed on SEN(B) specimens). Figure 10 shows the constant-amplitude tests plotting $\sigma_{max}$ (notch root elastic stress) against cycles to failure ($N_f$). The square symbols are single-edge-notch tension, SEN(T), specimens [44], whereas the solid circular symbols are from the current study. A trial-and-error procedure was used to find the 6-μm semi-circular surface flaw at the center of the notch to fit the fatigue data. Herein, the 6-μm flaw is considered an equivalent initial flaw size (EIFS) to fit the S-N behavior under constant-amplitude loading. The solid curve shows calculated lives that used available crack-growth-rate data (rates $\geq 4 \times 10^{-11}$ m/cycle) for $\sigma_{max} \geq 980$ MPa. The dashed (blue) curve used the estimated $\Delta K_{eff}$-rate relation for rates below $4 \times 10^{-11}$ m/cycle (see Figure 9). On Figure 10, the horizontal dashed (black) line is where $\Delta K_{eff} = (\Delta K_{eff})_{th} = 2.3$ MPa$\sqrt{m}$. Upper and lower bound calculations were made with 4- and 10-μm, respectively.

Figure 11 shows the test data (solid circular symbols) under the Cold-Turbistan+ spectrum. The open symbols are the retest of the two runout tests, but at higher applied notch elastic stress. FASTRAN predictions using the same 6-μm surface flaw fell at the lower bound of the test data using the baseline curve from Figure 9 (Table 1). If a threshold of 2.3 MPa$\sqrt{m}$ (no crack growth) was selected, then the cycles to failure were near the upper bound of the test data. Selecting a $\Delta K_{eff}$-rate relation between the vertical dashed line and the baseline (blue) curve below $10^{-10}$ m/cycle, would have given a more accurate predicted fatigue strength. These results indicate that the selection of the baseline curve in Figure 9 for rates below $10^{-10}$ m/cycle is very important. In addition, in the fatigue endurance-limit region, there could also be some build-up of fretting oxide debris on the small-crack surfaces, which could increase the fatigue life by elevating the crack-opening loads. Further study is required to include fretting debris on the crack surfaces in the FASTRAN model.

To study the crack-closure behavior and life predictions under the Cold-Turbistan+ spectrum, the predicted crack-opening-stress ratios are shown in Figure 12 for a small portion of the spectrum (first 80 cycles). Here, the FASTRAN code used the full crack-closure model with a constraint factor of 2.5. Calculated crack-opening stresses began at the minimum stress in the spectrum and the crack-opening stress increases as the crack grows and leaves plastically deformed material in the wake of the crack tip. Calculations were also made at Block 35 (about half-life) and show that the crack-opening stresses are, generally, higher than Block 1. These results are somewhat surprising, in that the predicted crack-opening stresses were generally at the minimum applied stress values for the larger cyclic amplitudes. This would imply that the loading cycle would be fully

effective in growing the crack. Thus, the Cold-Turbistan+ spectrum would be classified as an "accelerating" spectra.

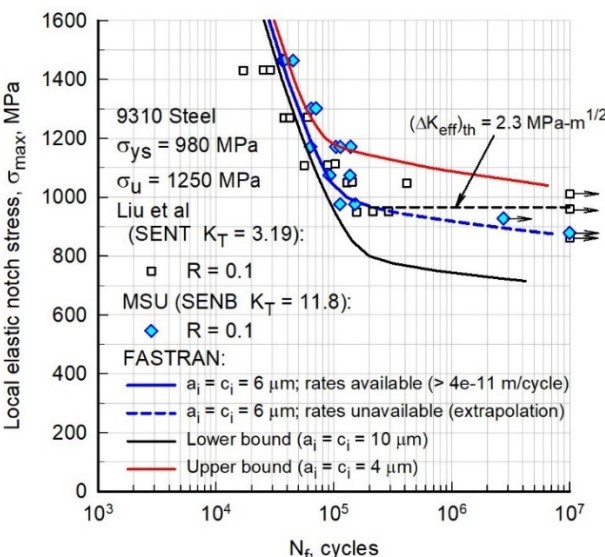

**Figure 10.** Measured and calculated stress-life behavior under constant-amplitude loading.

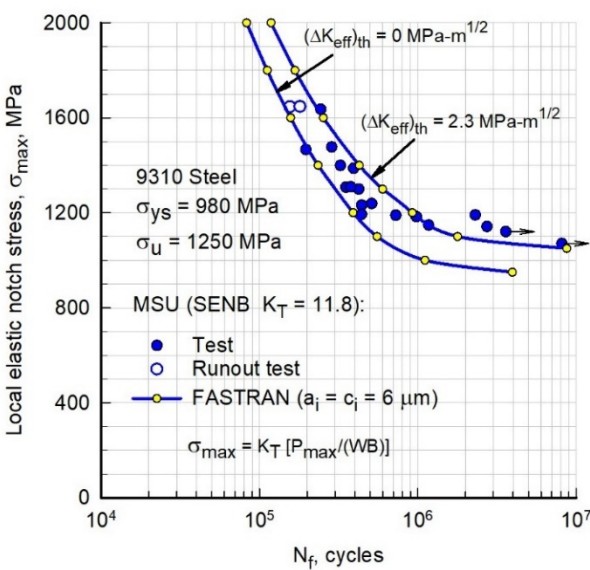

**Figure 11.** Measured and predicted stress-life behavior under Cold-Turbistan+ spectrum loading.

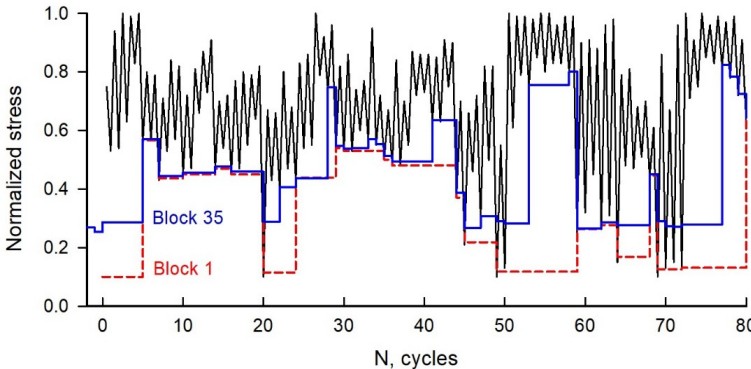

**Figure 12.** Part of Cold Turbistan+ spectrum loading and calculated crack-opening-stress ratios.

These results are important, in that Rainflow analyses for fatigue-crack growth are a function of load history. Fortunately, the current Rainflow analyses, used in the literature, would be conservative without considering crack-closure load-history effects. However, the accuracy of fatigue-crack-growth calculations under spectrum loading would be greatly improved with "Rainflow-on-the-Fly" methodology.

## 7. Concluding Remarks

Fatigue of materials, like alloys, is basically the fatigue-crack growth in small cracks nucleating at micro-structural features, such as inclusions and voids, or at micro-machining marks, and large cracks growing to failure. Thus, the traditional fatigue-crack nucleation stage ($N_i$) is basically the growth in microcracks (initial flaw sizes of 1 to 30 μm growing to about 250 μm) in a variety of metal alloys. Large-crack growth and failure are regions where fracture-mechanic parameters were successful in correlating and predicting fatigue-crack growth and fracture. In the last three decades, fracture-mechanics concepts have also been successful in predicting "fatigue" (growth of small cracks) under constant-amplitude and spectrum loading using crack-closure theory. Therefore, the crack-growth approach provides a unified theory for the determination of fatigue lives for metal alloys. However, for pure- and single-crystal materials, there are nucleation cycles required to transport dislocations at critical locations, develop slip bands, and cracks.

Tests were conducted on compact, C(T), and single-edge-notch-bend, SEN(B), specimens made of a 9310 steel ($B$ = 6.35 mm) under laboratory air and room temperature conditions. The C(T) crack-growth specimens were tested over a wide range in stress ratios ($R$ = 0.1 to 0.95) and crack-growth rates from threshold to fracture. A crack-closure model (FASTRAN) was used to collapse the $\Delta K$-rate data onto an almost unique $\Delta K_{eff}$-rate relation over more than four orders-of-magnitude in rates.

Fatigue tests were conducted on the SEN(B) specimens under constant-amplitude loading ($R$ = 0.1) and a Cold-Turbistan+ spectrum loading. The $\Delta K_{eff}$-rate crack-growth relation was used to calculate or predict fatigue behavior on the SEN(B) specimens using the crack-closure model and small-crack theory. The constant-amplitude fatigue tests were used to determine an initial semi-circular surface flaw size (6-μm) to fit the test data. The 6-μm flaw is considered an equivalent initial flaw size (EIFS). Using the same initial flaw size enabled the FASTRAN code to predict the fatigue behavior under the Cold-Turbistan+ spectrum loading quite well. Rainflow-on-the-Fly methodology was validated on a complex spectrum loading and indicated that the calculated damage was a function of load history and that the usual Rainflow methods would not capture correct crack-growth damage, unless the method was updated during crack-growth history.

**Funding:** This research received no external funding.

**Conflicts of Interest:** The author declares no conflict of interest.

## Nomenclature

| | |
|---|---|
| *a* | crack depth in thickness direction, mm |
| $a_i$ | initial crack depth in thickness direction, mm |
| *B* | specimen thickness, mm |
| *c* | crack length in width direction, mm |
| $c_i$ | initial crack length in width direction, mm |
| *D* | single-edge-notch diameter, mm |
| *da/dN* | crack-growth rate in depth direction, m/cycle |
| *dc/dN* | crack-growth rate in width direction, m/cycle |
| *E* | modulus of elasticity, GPa |
| *F* | boundary-correction factor |
| $F_n$ | boundary-correction factor based on net-section stress |

| | |
|---|---|
| $K$ | stress-intensity factor, MPa$\sqrt{\text{m}}$ |
| $K_F$ | elastic-plastic fracture toughness, MPa$\sqrt{\text{m}}$ |
| $K_{Ie}$ | elastic fracture toughness, MPa$\sqrt{\text{m}}$ |
| $K_{max}$ | maximum stress-intensity factor, MPa$\sqrt{\text{m}}$ |
| $K_T$ | elastic stress-concentration factor |
| $m$ | fracture toughness parameter |
| $N$ | number of cycles |
| $N_f$ | number of cycles to failure |
| $P$ | applied load, kN |
| $P_f$ | failure load, kN |
| $P_{max}$ | maximum applied load, kN |
| $P_{min}$ | minimum applied load, kN |
| $R$ | load ratio ($P_{min}/P_{max}$) |
| $S$ | applied remote stress, MPa |
| $S_{max}$ | maximum applied stress, MPa |
| $S_{min}$ | minimum applied stress, MPa |
| $S_o$ | crack-opening stress, MPa |
| $w$ | specimen width, mm |
| $\alpha$ | tensile constraint factor |
| $\Delta K$ | stress-intensity factor range, MPa$\sqrt{\text{m}}$ |
| $\Delta K_c$ | stress-intensity-factor range at failure, MPa$\sqrt{\text{m}}$ |
| $\Delta K_{eff}$ | effective stress-intensity factor range, MPa$\sqrt{\text{m}}$ |
| $(\Delta K_{eff})_{\text{th}}$ | effective stress-intensity factor range threshold, MPa$\sqrt{\text{m}}$ |
| $(\Delta K_{eff})_{\text{T}}$ | effective stress-intensity factor range transition, MPa$\sqrt{\text{m}}$ |
| $\Delta K_o$ | effective threshold as a function of R, MPa$\sqrt{\text{m}}$ |
| $\rho$ | plastic-zone size, mm |
| $\sigma_o$ | flow stress (average of $\sigma_{ys}$ and $\sigma_u$), MPa |
| $\sigma_{ys}$ | yield stress (0.2 percent offset), MPa |
| $\sigma_u$ | ultimate tensile strength, MPa |
| $\omega$ | cyclic plastic-zone size, mm |

## Abbreviations

| | |
|---|---|
| BFS | back-face strain |
| CPCA | compression pre-cracking constant amplitude |
| CPLR | compression pre-cracking load reduction |
| C(T) | compact (tension) specimen |
| SEN(B) | single-edge-notch (bend) specimen |
| SEN(T) | single-edge-notch (tension) specimen |

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
