# Peer review of "Fatigue and Crack Growth under Constant- and Variable-Amplitude Loading in 9310 Steel Using “Rainflow-on-the-Fly” Methodology"

_metals, doi:10.3390/met11050807_

Round 1

Reviewer 1 Report

This paper presents an investigation regarding the Fatigue and Crack Growth under Constant- and Variable-Amplitude Loading in 9310 Steel Using “Rainflow-On-The-Fly” Methodology. Based on correlations from FASTRAN life prediction code (subroutine “rainflow-on-the-fly”) and experimental results the obtained results are quite interesting.

In the reviewer`s opinion:

- the paper is interesting to the scope of the journal

- the subject addressed in this article is worthy of investigation

- the title is an accurate indication of the contents

- the abstract is adequate

- the organization of the manuscript is appropriate

- the written English is ok

The paper contains interesting results but should be subjected to minor improvements considering the following issues:

In page 3: line 105:  “The spectra were applied ...of 5 mm”. This specimen should also appear in Fig. 1.

Page 5, line 140 – check word “predicteded”

Page 7, line 179 – BFS...missing in nomenclature

Page 7, line 184 – Please develop a little more and clarify the reasons why exists spread in both fracture and threshold regions

Page 11, line 290 and line 293 – correct the units

Page 12, fig.11 – It seems that estimations from FASTRAN for higher fatigue lives are not so good...can Author develop a little more....

Author Response

Reviewer #1

This paper presents an investigation regarding the Fatigue and Crack Growth under Constant- and Variable-Amplitude Loading in 9310 Steel Using “Rainflow-On-The-Fly” Methodology. Based on correlations from FASTRAN life prediction code (subroutine “rainflow-on-the-fly”) and experimental results the obtained results are quite interesting.

The paper contains interesting results but should be subjected to minor improvements considering the following issues:

 In page 3: line 105:  “The spectra were applied ...of 5 mm”. This specimen should also appear in Fig. 1.

The Rainflow-on-the-Fly calculations were made for "a crack in an infinite plate under remote uniform stress, S".  Thus, a drawing of an infinite plate with a crack under remote applied stress is not necessary.  I think that the statement should be sufficient.

 Page 5, line 140 – check word “predicteded” 

Word spelling was corrected.

 Page 7, line 179 – BFS...missing in nomenclature  

BFS was added to nomenclature.  In addition, ΔKc was also added.

 Page 7, line 184 – Please develop a little more and clarify the reasons why exists spread in both fracture and threshold regions   

See sentences and 2 references added.

In the fracture region, the spread is due to fracture being controlled by the maximum stress-intensity factor and not by ΔK. For example, the stress-intensity-factor range at failure, ΔKc = KIe (1 – R), where KIe is the elastic fracture toughness. Thus, high R tests would fracture at a lower ΔKc value than low R tests. The spread in the threshold region is suspected to be due to load-history effects caused by the load-reduction procedure used in the CPLR test method. Load-reduction tests, as specified in ASTM E-647 [28], have been shown [25,26] to induce more spread in ΔK in the threshold region than the mid-rate (Paris) region. The spread has been associated with a rise in the crack-closure behavior during load-reduction tests [34,35].

  1. Yamada, Y.; Newman, J.C. Jr. Crack closure under high load-ratio conditions for Inconel 718 near threshold behavior. Engineering Fracture Mechanics; 2009; 76; pp. 209-220.
  2. Yamada, Y.; Newman, J.C. Jr. Crack closure behavior of 2324-T39 aluminum alloy near threshold conditions for high load ratio and constant Kmax tests. International Journal of Fatigue; 2009; 31; pp. 1780-1787.

 Page 11, line 290 and line 293 – correct the units   

Two Greek letters were missing and they were corrected.

 Page 12, fig.11 – It seems that estimations from FASTRAN for higher fatigue lives are not so good...can Author develop a little more...

At 1200 MPa, 4 tests had fatigue lives ranging from 42,000 to 2,200,000 cycles (factor of 52 difference), and at about 1100 MPa there were 2 runout tests.  For long life fatigue tests near the endurance limit there could be orders-of-magnitude differences in fatigue lives.  It is the fatigue strength that is important.  Using the lower predicted curve, the fatigue limit was predicted to be 15% low.

In Figure 9, selecting a DKeff-rate relation between the vertical dashed line and the baseline (blue) curve below 1e-10 m/cycle, would have given a more accurate predicted fatigue strength (like 8% low). These results indicate that the selection of the baseline curve for rates below 1e-10 m/cycle is very important. In addition, in the fatigue endurance-limit region, there could also be some build-up of fretting oxide debris on the small-crack surfaces, which could increase the fatigue life by elevating the crack-opening loads.

Reviewer 2 Report

The current study investigates the fatigue and crack behaviour over a wide range in rates from threshold to fracture for load ratios 1 to 0.95 in 9310 Steel materials subjected to constant and variable amplitude loading using Rainflow-On-The-Fly Methodology. Authors used FASTRAN used to develop the baseline crack-growth-rate curve and model was validated.

The abstract can be improved please answer the following in brief and concise manner: Please consider reviewing the abstract and highlight the novelty, major findings and conclusions.

The literature review is short and must be extended, the authors must provide a comprehensive literature review which covers previous studies similar to this work, report what they did and what were their main findings and how does your current work brings new knowledge and different to the field.

What is the research gap did you find from the previous researchers in your field? Mention it properly. It will improve the strength of the article.

What are the limitations in the FASTRAN subroutine compared to other similar codes used to model fatigue in metals?

Line 314 “These results are somewhat surprising” please explain further why you think these results are contradicting from what you expect to have and also relate them to past studies in the open literature and try to explain why there are different trends

Conclusion can be improved and expanded.

The results are merely described and is limited to comparing the experimental observation. The authors are encouraged to include more detailed discussion in each of the results section and critically discuss the observations from this investigation with existing literature.

Author Response

Reviewer #2

The current study investigates the fatigue and crack behaviour over a wide range in rates from threshold to fracture for load ratios 0.1 to 0.95 in 9310 Steel materials subjected to constant and variable amplitude loading using Rainflow-On-The-Fly Methodology. Authors used FASTRAN to develop the baseline crack-growth-rate curve and model was validated.

The abstract can be improved please answer the following in brief and concise manner: Please consider reviewing the abstract and highlight the novelty, major findings and conclusions.

The abstract has been modified with a few more statements on the novelty and major findings in the subject paper.

The literature review is short and must be extended, the authors must provide a comprehensive literature review which covers previous studies similar to this work, report what they did and what were their main findings and how does your current work bring new knowledge and different to the field.

The introduction has been greatly modified with several new paragraphs on a literature review (including a number of other references on the same subject matter) and more background on the approach presented in the subject paper.

What is the research gap did you find from the previous researchers in your field? Mention it properly. It will improve the strength of the article.

The author does not know of any previous researchers who have developed methods or codes to predict the crack-opening-load history of small cracks growing from a notch under spectrum loading.  This is a unique aspect of the presented work.  And to truly predict the “fatigue” behavior under variable-amplitude loading from only constant-amplitude crack-growth and fatigue data.  Classical fatigue concepts have difficulty with variable-amplitude loading.

What are the limitations in the FASTRAN subroutine compared to other similar codes used to model fatigue in metals?

The author has reviewed the literature and found two “review papers” on fatigue.  The treatment of fatigue under variable-amplitude loading by traditional methods require non-linear damage rules fitted to variable-amplitude tests.  Are these truly predictable methods?

Line 314 “These results are somewhat surprising” please explain further why you think these results are contradicting from what you expect to have and also relate them to past studies in the open literature and try to explain why there are different trends.

The author has removed the statement “These results are somewhat surprising …”.  For the Cold Turbistan+ spectra, the author would have expected higher crack-opening loads but several times in the past the FASTRAN code has predicted the opposite of what one may expect.  In my opinion, the FASTRAN algorithm models the load-history effects quite accurately.  Several years ago, Prof Schijve has some block-loading tests on a Russian (D16Cz) aluminum alloy that was opposite from what he would have expected.  But FASTRAN predicted the correct trend in the block-loading tests (Engineering Fracture Mechanics, Vol. 78, 2011, pp. 2609-2619). 

And the results from FASTRAN predicted the “fatigue life” under the Cold Turbistan+ spectrum using a flaw size determined from constant-amplitude tests on 9310 steel.  These results add credibility to the crack-opening loads predicted.

Conclusion can be improved and expanded.

The Concluding Remarks section was greatly modified to include background information, more details on the test program and key conclusions from current paper.

The results are merely described and is limited to comparing the experimental observation. The authors are encouraged to include more detailed discussion in each of the results section and critically discuss the observations from this investigation with existing literature.

More detailed discussions were included in the revised paper.

Reviewer 3 Report

The article summarizes some regularities of the cyclic crack resistance of materials and structures under сconstant- and variable-amplitude loading in 9310 steel. But in the introductory part, only the works of the author are analyzed, as well as the publications of prof. R. Pippan and others well-known ("classical") scientists.

I propose to supplement the analysis of publications with articles by other authors who:

- are the ideological followers of the ideas of Prof.’s Pippan, Newman, and others.

- developed these approaches in their publications.

I suggest analyzing for example:

Konovalov, S., Komissarova, I., Ivanov, Y., Gromov, V., Kosinov, D. Structural and phase changes under electropulse treatment of fatigue-loaded titanium alloy VT1-0 (2019) Journal of Materials Research and Technology, 8 (1), pp. 1300-1307. DOI: 10.1016/j.jmrt.2018.09.008;

Maruschak, P., Panin, S., Vlasov, I., Prentkovskis, O., Danyliuk, I. Structural levels of the nucleation and growth of fatigue crack in 17Mn1Si steel pipeline after long-term service (2015) Transport, 30 (1), pp. 15-23.  DOI: 10.3846/16484142.2014.1003404;

Pyndus, Y., Yasniy, O., Fostyk, V., Maruschak, P. Assessment of Minimal Fatigue Crack Growth Rate After a Single Overload in D16chT Alloy (2018) Iranian Journal of Science and Technology - Transactions of Mechanical Engineering, 42 (4), pp. 341-346. DOI: 10.1007/s40997-017-0101-5

Author Response

Reviewer #3

The article summarizes some regularities of the cyclic crack resistance of materials and structures under сonstant- and variable-amplitude loading in 9310 steel. But in the introductory part, only the works of the author are analyzed, as well as the publications of prof. R. Pippan and others well-known ("classical") scientists.

The abstract has been modified with a few more statements on the novelty and major findings in the subject paper. The introduction has been greatly modified with several new paragraphs on a literature review (including a number of other references on the same subject matter) and more background on the approach presented in the subject paper. The Concluding Remarks section was greatly modified to include background information, more details on the test program and key conclusions from current paper.

I propose to supplement the analysis of publications with articles by other authors who:

- are the ideological followers of the ideas of Prof.’s Pippan, Newman, and others.

- developed these approaches in their publications.

I suggest analyzing for example:

Konovalov, S., Komissarova, I., Ivanov, Y., Gromov, V., Kosinov, D. Structural and phase changes under electropulse treatment of fatigue-loaded titanium alloy VT1-0 (2019) Journal of Materials Research and Technology, 8 (1), pp. 1300-1307. DOI: 10.1016/j.jmrt.2018.09.008;

Maruschak, P., Panin, S., Vlasov, I., Prentkovskis, O., Danyliuk, I. Structural levels of the nucleation and growth of fatigue crack in 17Mn1Si steel pipeline after long-term service (2015) Transport, 30 (1), pp. 15-23.  DOI: 10.3846/16484142.2014.1003404;

Pyndus, Y., Yasniy, O., Fostyk, V., Maruschak, P. Assessment of Minimal Fatigue Crack Growth Rate After a Single Overload in D16chT Alloy (2018) Iranian Journal of Science and Technology - Transactions of Mechanical Engineering, 42 (4), pp. 341-346. DOI: 10.1007/s40997-017-0101-5

The reviewer suggested that the author conduct analyses on some of these 3 papers.  It is beyond the scope of the present paper to analyze results from the 3 listed papers.

Round 2

Reviewer 2 Report

All questions answered